# A Core Module of Nuclear Genes Regulated by Biogenic Retrograde Signals from Plastids

**DOI:** 10.3390/plants10020296

**Published:** 2021-02-04

**Authors:** Björn Grübler, Carolina Cozzi, Thomas Pfannschmidt

**Affiliations:** Pflanzenphysiologie, Institut für Botanik, Naturwissenschaftliche Fakultät, Leibniz-Universität Hannover, Herrenhäuser Str. 2, 30419 Hannover, Germany; BjoernGruebler@gmx.de (B.G.); c.cozzi@botanik.uni-hannover.de (C.C.)

**Keywords:** plastids, photomorphogenesis, retrograde control, biogenic signals, lincomycin, norflurazon, *pap7-1* mutant

## Abstract

Chloroplast biogenesis during seedling development of angiosperms is a rapid and highly dynamic process that parallels the light-dependent photomorphogenic programme. Pre-treatments of dark-grown seedlings with lincomyin or norflurazon prevent chloroplast biogenesis upon illumination yielding albino seedlings. A comparable phenotype was found for the *Arabidopsis* mutant *plastid-encoded polymerase associated protein 7* (*pap7*) being defective in the prokaryotic-type plastid RNA polymerase. In all three cases the defect in plastid function has a severe impact on the expression of nuclear genes representing the influence of retrograde signaling pathway(s) from the plastid. We performed a meta-analysis of recently published genome-wide expression studies that investigated the impact of the aforementioned chemical and genetic blocking of chloroplast biogenesis on nuclear gene expression profiles. We identified a core module of 152 genes being affected in all three conditions. These genes were classified according to their function and analyzed with respect to their implication in retrograde signaling and chloroplast biogenesis. Our study uncovers novel genes regulated by retrograde biogenic signals and suggests the action of a common signaling pathway that is used by signals originating from plastid transcription, translation and oxidative stress.

## 1. Introduction

Chloroplasts are sub-cellular organelles in plants and algae that perform photosynthesis and many other metabolic activities. In angiosperms they develop from undifferentiated precursors called proplastids which are inherited from the mother plant in the cells of the embryo [1,2]. Upon germination, the embryo develops into a rapidly growing seedling. In light this development follows a photomorphogenic programme which includes opening of the cotyledons, repression of hypocotyl elongation and greening. The latter is due to the biogenesis of chloroplasts from the proplastids in the cotyledons. However, in the case that germination occurs in the dark because the seed is buried by humus or soil, the seedling follows a different developmental programme called skotomorphogenesis [3]. Here, the cotyledons remain small and of yellow color without any expansion. They are directed downwards by an apical hook which protects the apical meristem while the highly elongating hypocotyl drives the cotyledons towards the soil surface. The proplastids in this etiolated seedling develop into yellow etioplasts, an intermediate developmental stage of plastids incapable of performing photosynthessis. However, etioplasts develop into chloroplasts within hours as soon as the germinating seedling perceives light. Etioplast-to-chloroplast conversion, thus, is often used as experimental system to study molecular basics of chloroplast development [4,5,6,7].

The molecular steps controlling the biogenesis of chloroplasts are far from understood, mostly because of the rapidity and complexity of the processes involved [7]. Initiation of photomorphogenesis, starting either directly from proplastids or from etioplasts, occurs by the activation of the phytochrome system through light. It largely determines the morphological changes of the seedling when it enters photomorphogenesis. Chloroplast biogenesis occurs at the same time; however, many observations in recent years indicate that this developmental process is rather a parallel than an intrinsic part of photomorphogenesis [8]. For instance *constitutive photomorphogenesis* (*cop*) mutants from *Arabidopsis* develop a photomorphogenic phenotype in the dark without chloroplast formation [9]. Vice versa, *plastid-encoded RNA polymerase-associated protein* (*pap*) mutants develop a normal photomorphogenic phenotype in the light without chloroplast formation [10]. Chloroplast biogenesis, therefore, is neither a prerequisite nor a consequence of photomorphogenesis and it remains to be elucidated how it is connected to the photomorphogenic programme.

The major steps of chloroplast biogenesis involve the build-up of the internal thylakoid membrane system and the assembly of the photosynthetic apparatus. Because of their endosymbiotic ancestry chloroplasts possess their own genome that encodes central components of the photosynthetic and gene expression machineries [11]. However, assembly of functional membrane structures and protein complexes requires the import of thousands of nuclear-encoded components. Plastids, thus, are regarded as genetically semi-autonomous. Because of the high copy-number of the plastid genome and the fact, that each cell contains many plastids, one can observe a strong imbalance in the ratio of plastid over nuclear genes encoding plastid proteins. Proper timely and spatial expression of genes essential for chloroplast biogenesis, therefore, requires a high coordination between the two genetic compartments. This is achieved by a mutual information exchange called anterograde signaling (nucleus-to-plastid signaling) and retrograde signaling (plastid-to nucleus signaling) [12,13,14,15,16,17].

Retrograde signals from plastids during early steps of chloroplast biogenesis have been named biogenic signals (in contrast to operational signals from fully active chloroplasts) [18]. These signals were discovered in experiments where plastid development was either chemically or genetically inhibited resulting in a parallel inhibition of the expression of nuclear genes encoding plastid photosynthesis proteins such as subunits of the ribulose-bis-phosphate carboxylase/oxygenase (Rubisco) or light harvesting complexes (Lhc) of the photosystems [19,20]. This led to the concept of a plastid factor or signal that is required for the proper development of the chloroplast [21]. This research field has seen a tremendous effort in the last decade and it became clear that biogenic signals from plastids likely play an important role in the regulation of chloroplast biogenesis. How they are implicated in detail, is however, far from understood [22].

Two common approaches for the study of biogenic signals are the treatments of germinating seedlings with norflurazon (NF) and lincomycin (LIN). NF is an inhibitor of the plastid phytoene desaturase and a potent repressor of carotenoid biosynthesis. NF-treated dark-grown seedlings remain completely white and experience a severe oxidative stress from photo-sensitization of protochlorophyllide upon illumination since the quenching properties of the carotenoids are missing. The generated reactive oxygen species block any further steps towards chloroplast biogenesis by oxidative destruction of the internal plastid structures [23]. LIN treatment, in contrast, blocks the plastid translation machinery and prevents the build-up of all plastid-encoded protein components of the photosynthesis apparatus including the core proteins of the photosystems which are essential for a proper assembly of the systems [24]. Both treatments act on different sites in the plastid, but the resulting phenotypes both at phenotypic and molecular levels are similar in many aspects including an albino appearance and a repression of nuclear encoded photosynthesis associated nuclear genes (PhANGs).

Recently we reported a detailed expression profiling of the *pap7-1* mutant of *Arabidopsis* [25]. This mutant displays a defect in the activity of the plastid encoded RNA polymerase (PEP) and exhibits a severe disturbance in plastid and nuclear gene expression leading to an albino phenotype. Surprisingly, the repressive impact on PhANG expression in this mutant was rather weak and strong repression was found to be limited to the group of *Lhc* genes contrasting the notion that inhibition of plastid development causes a general repression of PhANGs. Therefore, we were wondering what the “true” impact of biogenic signals on chloroplast biogenesis is. Here, we present a meta-analysis of gene expression profiles obtained from the *Arabidopsis pap7-1* mutant and two recent gene expression studies using NF and LIN in order to define commonalities and differences between the three approaches. Our data uncover a core gene module that exhibits common expression profiles in all three conditions and identify potential novel targets of biogenic signals as well as regulators of chloroplast biogenesis.

## 2. Material and Methods

### 2.1. Sets of Retrograde Controlled Genes Used in This Study

Primary expression data sets for genes being controlled by retrograde signals identified under influence of NF and LIN treatments were taken from previously published studies [26,27]. Data sets for retrograde controlled genes in the *pap7-1* mutant as well as for light-controlled genes in wild-type plants were taken from our own data sets [25]. The study on LIN effects was performed as a kinetic experiment with samples taken 0.5, 1, 4 and 24 h after a light intensity shift. In pre-selection comparisons we observed that the number of overlapping genes between the *pap7-1* and LIN data sets increased with time. The 24 h LIN data set, therefore, was used as base for further meta-analysis. Only genes that exhibited a significant relative expression change (repression or induction) in response to plastid dysfunction of at least log2 ≥ 1 and a *p*-value ≤ 0.05 in all data sets were included in our comparisons. 

### 2.2. Comparison of Gene Lists

For all principal comparisons of microarray data sets and the identification of overlapping genes between the three studies we used standard functions of Microsoft EXCEL. The corresponding Venn diagram was generated by using a web-based tool for comparison of large data sets (InteractiVenn; http://www.interactivenn.net/index.html) [28].

### 2.3. Functional Annotation and Localization

The initial functional annotation of the identified 152 genes is based on the MapMan Bin categories [29] and was curated manually for gene descriptions and functions of encoded proteins afterwards. To this end each gene was checked for database entries in The Arabidopsis Information Resource (TAIR) (https://www.arabidopsis.org) and The Universal Protein Resourcce (UniProt) (https://www.uniprot.org). Information on potential interaction partners was obtained from the Biological General Repository for Interaction Datasets (BioGRID) (https://thebiogrid.org) and the real-time multiple association network integration algorithm for predicting gene function (GeneMANIA) (https://genemania.org) databases. In addition, information on potential localization of gene products was extracted from the Bio-Analytic Resource (BAR) for Plant Biology and the integrated Arabidopsis Cell eFP Browser (https://bar.utoronto.ca) and aligned with the corresponding information in the Arabidopsis thaliana chloroplast protein database AT_CHLORO (http://at-chloro.prabi.fr) for sub-plastidial localization [30]. Genes were then classified according to major functional categories (which may differ in some cases from the original MapMan bins) and were given as heat map sorted on the base of the expression values in the LIN data set. All collected information about the gene products are summarized in Appendix A.

## 3. Results and Discussion

### 3.1. Selection and Comparability of Microarray Data Sets

Recently, we identified a set of retrograde controlled genes that are repressed or induced by biogenic signals in the light when chloroplast biogenesis is blocked at the level of plastid transcription [25]. We were interested in understanding how similar or different these gene groups are in comparison to conditions when the block of chloroplast biogenesis occurs at the level of plastid translation or at a level of general destruction through oxidative damage. To this end we compared our data with results from two investigating the effects of LIN and NF on nuclear gene expression [26,27]. The experimental design of all three studies was highly similar using *Arabidopsis* seedlings grown for 5–7 days on sugar-supplemented medium in Petri dishes and exposed to light to visualize the effect of blocked chloroplast biogenesis on light-regulated gene expression profiles. In LIN-based experiments seedlings were grown in absence and presence of 0.5 mM LIN under extremely weak blue-red light of 0.5 µE fluence rate for 6 days followed by a shift to 60 µE blue-red light for 24 h [27]. In the NF experiment seedlings were grown in absence and presence of 5 µM NF for 3 days in the dark followed by 3 days in white light [26]. In both studies the inhibitors, thus, had sufficient time to block the respective process before induction of chloroplast biogenesis. In our own experiments with the genetically blocked *pap7-1* mutant seedlings were grown for 5 days directly under light since the genetic inactivation had become effective already during the establishment of the seeds [25]. Therefore, in all three experiments the effect of a block in chloroplast biogenesis on light-controlled gene expression was studied at a comparable developmental stage of the seedlings (2 cotyledons stage). Furthermore, all studies used the Affymetrix ATH1 *Arabidopsis* chips providing high technical comparability. We, therefore, regarded the experimental set-ups as sufficiently similar to provide reliable data for a meta-analysis.

### 3.2. Identification of a Core-Module of Genes Controlled by Biogenic Signals

The chosen gene sets contained 3004 genes for the LIN set, 1957 genes for the *pap7-1* set and 1128 genes for the NF set. Comparison of the three conditions revealed a common gene module of 152 genes being significantly regulated in all three conditions (Figure 1A). The highest bivalent similarity was found for the NF set that shared 51.1% of its genes with the LIN set and 21.5% with the *pap7-1* set. The *pap7-1* set shared 25.8% of its genes with the NF set and 12.4% with the LIN set while the LIN set shared 19.2% of its genes with the NF set and 16.8% with the *pap7-1* set (Figure 1B). These results strongly suggest that each of the three different blocks in chloroplast biogenesis induce largely their own specific responses in nuclear gene expression. These differences in expression are likely attributed to the different sites of the respective blocks and/or the slightly different experimental set-ups and laboratory conditions. Nonetheless, there apparently exists also a common and robust element in all three conditions giving rise to the shared group of 152 genes. Therefore, we regard this group as a highly trustful core module of genes being influenced in their expression when chloroplast biogenesis is blocked, thus representing genuine target genes regulated (directly or indirectly) by retrograde biogenic (RB) signals.

An initial functional annotation of the 152 genes based on the bin definition in MapMan [31] was performed in order to understand the regulatory implications of the biogenic control. The largest group comprised 38 genes encoding proteins of unassigned, unknown or hypothetical function (Figure 1C) strongly suggesting that many aspects of chloroplast biogenesis at the molecular level are not understood yet. The second largest group contained 18 genes encoding components implicated in “Photosynthesis” followed by groups involved in Protein” (15), “RNA” (11), “Redox” (10), “Transport” (9), “Hormone metabolism” (6), “Stress” (5) and 14 further groups represented by one to four genes.

The expression behavior of the 152 genes was highly similar in the three analyzed conditions with approximately one third (47) of genes being stronger expressed than in unaffected wild-type (WT) controls and two third (105) of genes exhibiting a weaker expression (Appendix A). In control conditions detecting the influence of light (WT-light versus WT-dark) most of the genes displaying a weaker expression upon the block in chloroplast biogenesis exhibited an opposite response with a clear induction upon illumination (and vice versa). This opposite expression pattern corresponds to recent reports where this pattern was interpreted in a way that biogenic signals turn light signals from positive into negative stimuli [32]. We, therefore, compared the core module to a set of recently identified direct HY5 target genes [33]. HY5 is one of the central regulators in photomorphogenesis and expected to control or co-regulate a large part of the light regulated genes [34] during seedling photomorphogenesis and chloroplast biogenesis. However, we identified only 5 of the 152 genes in the set of HY5-dependent genes (Appendix A) indicating that the core module is likely not directly regulated by the classical light-dependent signaling pathways.

Except for a few condition-specific expressed genes (see below), the general profiles of genes affected by LIN and NF treatments were found to be highly similar in strength and direction of expression change. The expression profiles in the *pap7-1* mutant were comparable to both LIN and NF profile concerning the direction of expression change; however, the degree of variation was weaker in most cases. Pharmaceutical inhibition of chloroplast development by LIN or NF treatments, thus, appears to have a stronger impact on nuclear gene expression than the genetic block in the *pap7-1* mutant, although the overall plant phenotype in all three conditions was largely the same. This suggests that the inhibitor treatments (i) induce stronger inhibitory effects in the plastid than the genetic block and/or (ii) the respectively affected plastid processes contribute to biogenic retrograde signaling with different strength. This suggest that the albino phenotype *per se* is likely not the cause of the changes in nuclear gene expression arguing for a specific molecular signaling pathway.

### 3.3. Functional Subsets within the Core Module

Because of the high number of genes encoding proteins with unknown or unassigned functions, we curated the functional annotation of each gene in the identified core module manually using current databases and literature and generated an up-dated list that covers detailed information on function, potential interaction partners and intracellular localization (Appendix A). By this means we uncovered a number of interesting novel targets for RB signals not yet described (see description of functional subsets below).

In total 84 of the 152 genes encode proteins that were predicted to be localized with low, medium or high probability in plastids. The plastid localization of 67 of them was experimentally confirmed by proteomic approaches according to AT_CHLORO. Roughly 45–55% of all genes in the module encode components that have predicted cellular destinations other than plastids (i.e., cytosol, Golgi apparatus, nucleus, endoplasmic reticulum, plasma membrane and mitochondria). Since prediction is not 100% precise and dual localization may also occur the results must be taken with care. Nevertheless, the data suggest that RB signals affect also other parts of the cell besides plastids. We noted that 80% of the genes with enhanced expression upon blocked chloroplast biogenesis belong to the group of genes encoding non-plastid localized components. Vice versa, 94% of the genes encoding plastid localized components displayed a low expression upon blocked chloroplast biogenesis indicating that the genes within the identified core module are dominated by two opposing expression modes that are largely associated with the potential localization of the affected gene products. In order to test whether or not the opposing expression modes are also associated to specific functions we analyzed the available information in a number of databases and ordered the genes of the core module into functional subsets with at minimum four genes (Figure 2A–F) and analyzed the corresponding expression profiles (see below for more details). 23 genes remained uncategorized and are given in the supplement (Appendix A)

### 3.4. Photosynthesis

In current literature photosynthesis-associated nuclear genes (PhANGs) are referenced as the “classical” target of biogenic retrograde signals [24,35]. Our meta-analysis indicates that PhANGs represent only a part of the gene groups being affected when chloroplast biogenesis is blocked (Figure 1C). Nevertheless, this group appears to be the one exhibiting the highest homogeneity in their mode of expression change among all groups of the module (Figure 2A–G) displaying a very low expression when chloroplast biogenesis is blocked while being highly induced upon etioplast-to-chloroplast conversion (WT-light vs. WT-dark). Interestingly, we identified only genes for components of the photosystem I (PSI) complex and for peripheral parts of photosystem II (PSII) including the light-harvesting antenna and the water splitting complex. Genes for subunits of the ATP-synthase, the Cyt*b_6_f*-complex and the NAD (P) H-dehydrogenase-like (NDH) complex as well as for enzymes of the Calvin-Benson cycle (with the phosphoribulokinase as only exception) were not identified. This implies a model in which RB signals modulate gene expression of PhANGs in a subset-specific way rather than a common overall control. It appears that especially components of the chlorophyll containing complexes respond to RB signals. Concomitant with this we observed a significant impact of RB signals on genes for key enzymes of Chl and carotenoid biosynthesis suggesting that RB signals may coordinate the syntheses of pigments and pigment-containing complexes.

### 3.5. Carbohydrate Metabolism and Transport

The direct products of photosynthesis are ATP and NADPH that are primarily used in the Calvin cycle to generate carbohydrates. However, they are also used in many other metabolic pathways that reside either partially or entirely in the plastid. Inhibition or down-regulation of photosynthesis, therefore, has a profound impact on the overall metabolic capacities of plastids. This is reflected in the effect of RB signals on the expression of numerous genes encoding plastid and cytosolic enzymes involved in carbohydrate metabolism that exhibit the same expression profile as the photosynthesis genes. Only the gene *GOLS1* (encoding the galactinol-synthase 1) exhibits an opposite expression profile. *GOLS1* has been proposed as a negative regulator of seed germination [36] as well as a gene responsive to various stressors. It is, thus, conceivable that RB signals from blocked chloroplast biogenesis indicate an early developmental plastid stage which is similar to that of proplastids in seeds or etioplasts of dark grown seedlings. This signal potentially arrests the gene expression programme that normally occurs when a seedling is exposed to light. In line with this assumption is the observation that a number of genes for transporter proteins required for the metabolic exchange across the chloroplast envelope show low expression. This accounts also for some transporters outside the plastid that are required to establish metabolic pipelines that end in a plastid (e.g., such as the nitrate transporter).

### 3.6. Redox Regulation

Many enzymatic reactions in chloroplasts are regulated via the redox state of the corresponding proteins. This control is exerted by a number of redox mediators such as thioredoxins that derive their reduction power from photosynthesis [37]. Many of the genes in this subset exhibit an expression profile similar to those within the photosynthesis module. The same is observed for genes encoding components of the antioxidant network (such as the gene for the dehydroascorbate reductase). All of them encode plastid localized proteins. Different expression profiles have been detected for several genes for cytosolic glutaredoxins that show strong accumulation. Many of these glutaredoxins are known to be involved in pathogen defense or stress responses [38,39]. Also other subsets contain components of the pathogen and stress response system suggesting that it is a major functional target of RB signals.

### 3.7. Development

As novel highly interesting targets for RB signals we identified three of the four genes encoding CURT (Curvature thylakoid) proteins within the core module. These proteins are required for the formation of the grana margins in thylakoid membrane system [40,41] and, therefore, absolutely essential for the build-up of the photosynthetic apparatus. Another novel target gene encodes SPD1 (seedling plastid development 1), a components required for eoplast de-differentiation and, therefore, being involved in very early developmental processes of chloroplast biogenesis [42]. All exhibit an expression profile corresponding to the photosynthesis subset. The same expression pattern was observed for the gene for peroxin 11, a component required for the build-up of peroxisomes [43]. Peroxisomes are functionally tightly coupled to chloroplasts as part of the photorespiratory pathway, but also in the synthesis of jasmonates in pathogen and wound response. Interestingly, an opposite expression pattern was observed for the gene encoding ACT8 (actin 8), a protein presumably involved in chloroplast movement or positioning suggesting that proper positioning may be required in early plastid biogenesis [44]. More general components involved in various aspects of plant development were also affected by RB signals; however, they exhibit expression patterns that are difficult to interpret. Only one gene in this subset encodes a plastid-localized protein. This protein is SEN1 (senescence 1), a component that is associated with senescence and strongly induced in the dark (therefore also known as dark-inducible 1) and upon phosphate starvation and biotic stresses [45]. It represents the gene with the strongest opposing expression between LIN and NF treatments (Appendix A). It was also observed to be strongly induced by abscisic acid (ABA) [46]. Since NF is blocking carotenoid production, and hence the ABA precursors, the opposing expression pattern is likely caused by this inhibitor-specific difference.

### 3.8. Transcription

Special interest in retrograde signaling is paid to the identification of nuclear localized transcription factors that mediate the corresponding gene expression responses (Figure 2E). An often discussed prime candidate, ABI4 (ABA insensitive 4), has been recently shown to be likely not involved in retrograde signaling [47]. Our meta-analysis identified nine genes encoding other potential candidates that exhibit diverse expression patterns. One highly interesting candidate is GLK1 that exhibits the “photosynthesis-type” transcript accumulation. It was shown to be a major activator of PhANG expression and chloroplast biogenesis [48] and is also down-regulated at protein level in *pap8-1*, an albino mutant related to *pap7-1* [49]. The low expression of GLK1 in response to blocked chloroplast biogenesis likely accounts for a large part of the expression profiles observed here. We noticed that the partner regulator GLK2 did not appear in our gene list suggesting that it is regulated differently. GLK1 may potentially work together with another candidate, COL7, a transcription factor involved in light signaling that interacts with HY5 (Appendix A). The gene for the MYB-domain containing transcription factor MYB29 shares the “photosynthesis-type” expression pattern. It is known to be a major activator of the biosynthesis of glucosinolates, secondary metabolites involved in responses to biotic and abiotic stresses [50]. This observation aligns well with the expression patterns of MAM1 and BGL28 (encoding two enzymes required for glucosinolate biosynthesis) (Appendix A). MYB29 is also involved in retrograde control from mitochondria regulating the expression of alternative oxidase 1a providing an interesting link between the two organelles [51]. Genes for three transcription factors RVE2, CGA1 and CRF6 (Reveille 2, Cytokinin-responsive GATA factor 1 and Cytokinin response factor 6) exhibit enhanced expression upon block of chloroplast biogenesis. Reveille 2, also a MYB-domain containing factor, is known to promote primary dormancy, but is repressed under illumination and imbibition by PhyB [52]. Its opposite expression in the NF sample may be an ABA-mediated effect as with SEN1 (see above). CGA1 and CRF6 are both implicated in cytokinin signaling. Cytokinin activates chloroplast biogenesis by inducing multiple target genes, among them HY5, CGA1 and CRF6. CGA1 is also known to act in an additive manner to GLK1 and both together represent a major regulatory hub of chloroplast biogenesis with GLKs likely acting down-stream of GNCs [53,54]. A recent study revealed that *pap* mutants overproduce cytokinins [55]. We regard it as likely that blocking of chloroplast biogenesis by LIN or NF induce similar responses. In sum, we identified a set of key transcription factors that is most likely involved in the mediation of RB signals during early chloroplast development and provides interesting candidates for future research in this context. Apart from directly affecting the expression of these key transcription regulators, RB signals appear to affect also other components involved in transcription or RNA metabolism both in plastid and in nucleus. A number of recent studies have identified numerous connections between retrograde signals and RNA metabolism in nucleus and plastid pointing to a potentially important regulatory level [56,57,58,59]. 

### 3.9. Proteins and Stress

Our meta-analysis was focused on the implications of RB signals on transcript accumulation and cannot account for potential effects on other gene expression levels. However, we observed significant effects of RB signals on genes encoding components involved in protein folding, proteostasis and stress responses, which imply that blocking chloroplast biogenesis affects also these functions. Recent studies revealed a major impact of plastid signals on the unfolded protein response in plastids, endoplasmic reticulum and cytosol as well as on protein accumulation [60,61,62]. This includes the action of heat shock proteins and likely other functionally related proteins such as immunophilins. The GLK1 transcription factor has been recently shown to be controlled at the protein level by RB signals [63] providing an important example. This aspect of retrograde signaling certainly will likely expand a lot in future research.

### 3.10. Lipids and Hormones

Lipids are major constituents of membranes and are absolutely essential for the build-up of chloroplast. Although blocking of chloroplast biogenesis results in plastids without thylakoid formation we did not find any impact on genes encoding proteins involved in plastid lipid metabolism, but only for non-plastid components. The corresponding gene expression profiles are very complex and difficult to interpret and more analyses will be necessary to understand and to explain the RB impact on them. We also found an impact of RB signals on multiple genes involved in hormone biosynthesis and/or signaling. We identified already in other functional subsets a number of candidate genes that are associated with synthesis and/or the action of several hormones (see above). ABA appears to be an interceptive signal in a few cases (Appendix A), but likely is not a major contributor to RB signaling in chloroplast biogenesis since NF treatment results in the same expression pattern as in the LIN and *pap7-1* profiles. 

Interestingly, we found a strong impact on two genes encoding key enzymes of allene oxide biosynthesis (lipoxygenase and allene oxide synthase) that produce the precursor for peroxisomal jasmonic acid production, the oxophytodienoic acid (OPDA) [64]. A distinct role of OPDA in retrograde signaling has been not yet reported, but the molecule represents a likely candidate for a metabolite signal since it leaves the plastid for further metabolic processing. The impact on the allene oxide pathway probably is connected to the missing allene oxide precursor molecule linolenic acid that usually originates from the thylakoid lipids and which are not present in the arrested plastid [64]. A number of other genes encoding components involved in or connected to plant defense to biotic and abiotic stressors or peroxisome biogenesis were identified in this study (see above) supporting the view that a block of chloroplast biogenesis generates a situation of severe stress that is not only based on the missing photosynthetic function, but also caused by a dysregulation in the build-up of the plant defense system. Besides, most other genes identified in the “Hormones” subset encode proteins with non-plastidial locations demonstrating the broad impact of RB signals on the hormone-signaling network. 

## 4. Conclusions

The core module responsive to RB signals identified in this study is different from retrograde controlled gene modules identified in earlier studies [65,66,67], most likely because these studies included conditions in which also retrograde operational (RO) signals are active (i.e., in the presence of fully developed chloroplasts). All three conditions analyzed in this study did not include the action of RO signals. The molecular targets causing the arrest in chloroplast biogenesis were different in the three conditions (cf. introduction), but the affected processes are not independent from each other and are interlinked by negative feedback loops affecting the generation of components of the transcriptional (rpo subunits) and translational (ribosomal components) machineries (Figure 3). Because of these connections and the high similarity in the three expression profiles we regard it as very likely that all three conditions generate signals that feed into the same signaling pathway(s). This signaling pathway targets mainly genes for photosynthesis or processes coupled to photosynthesis. Typically such genes are up-regulated by illumination and RB signals appear to intercept this light-dependent activation. A much smaller group of genes displayed the opposite expression profile indicating that RB signals can be of positive or negative effect. This master expression switch accounts for most genes in the core module. Only a few genes displayed more complex patterns that suggest the involvement of additional regulatory signals in their expression (such as ABA mediated signals). It must be noted that our meta-analysis describes only gene expression changes. Thus, it cannot draw conclusions on the positive or negative nature of RB signals and cannot distinguish between missing activation or active repression (or vice versa) [21,68]. It also cannot explain whether the listed genes are direct or indirect targets. We regard it as likely that most genes in this module are controlled by just a few regulators primarily targeted by transcription factors responding to the RB signal(s) (Figure 2E). This would explain the homogeneous expression profiles of so many different genes. Novel prime candidates for the mediation of the RB signal(s) towards such primary regulators are dually localized PEP-associated proteins (PAPs) that have been shown to be essential for chloroplast biogenesis and formation of late photobodies during early steps of photomorphogenesis [49,69,70]. They belong to a group of proteins that appear to move via the plastid towards the nucleus representing genuine retrograde signals [71]. The core module identified here provides novel insights into the targets of RB signals and may serve as base for more detailed working models in future studies. This will include the determination of the accumulation of the corresponding proteins.

## Figures and Tables

**Figure 1 plants-10-00296-f001:**
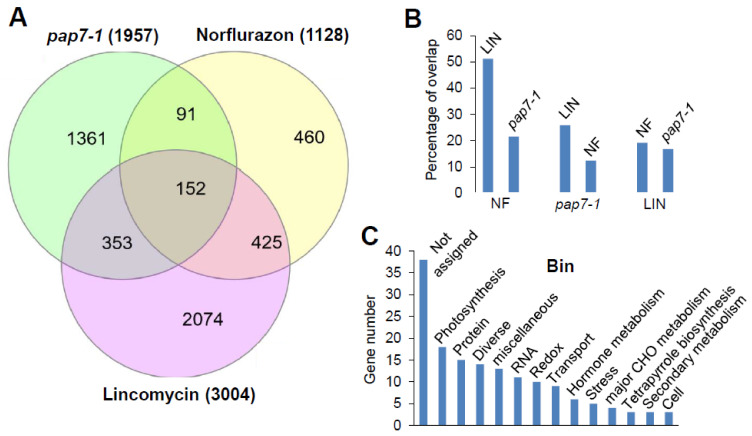
Comparison of target gene modules identified in three experimentally different approaches blocking chloroplast biogenesis. (**A**) Venn diagram describing the overlap in target genes of biogenic signals identified by norflurazon (NF) treatment, 24 h lincomycin (LIN) treatment and by genetic inactivation in *pap7-1* mutants. In all studies significantly regulated genes were defined by an expression change of at least log2 fold change ≥ 1 and a p-value ≤ 0.05. (**B**) Percentage of shared genes between the treatments. Since the sizes of the affected gene groups in the three conditions are different, the overlap is given for each of the three binoms (indicated on bottom) separately. (**C**) Functional annotation of genes shared between all three conditions. The categorization followed the Bin system of MapMan. Gene groups with a minimum of at least three genes are mentioned. A complete list of all genes is given in Appendix A.

**Figure 2 plants-10-00296-f002:**
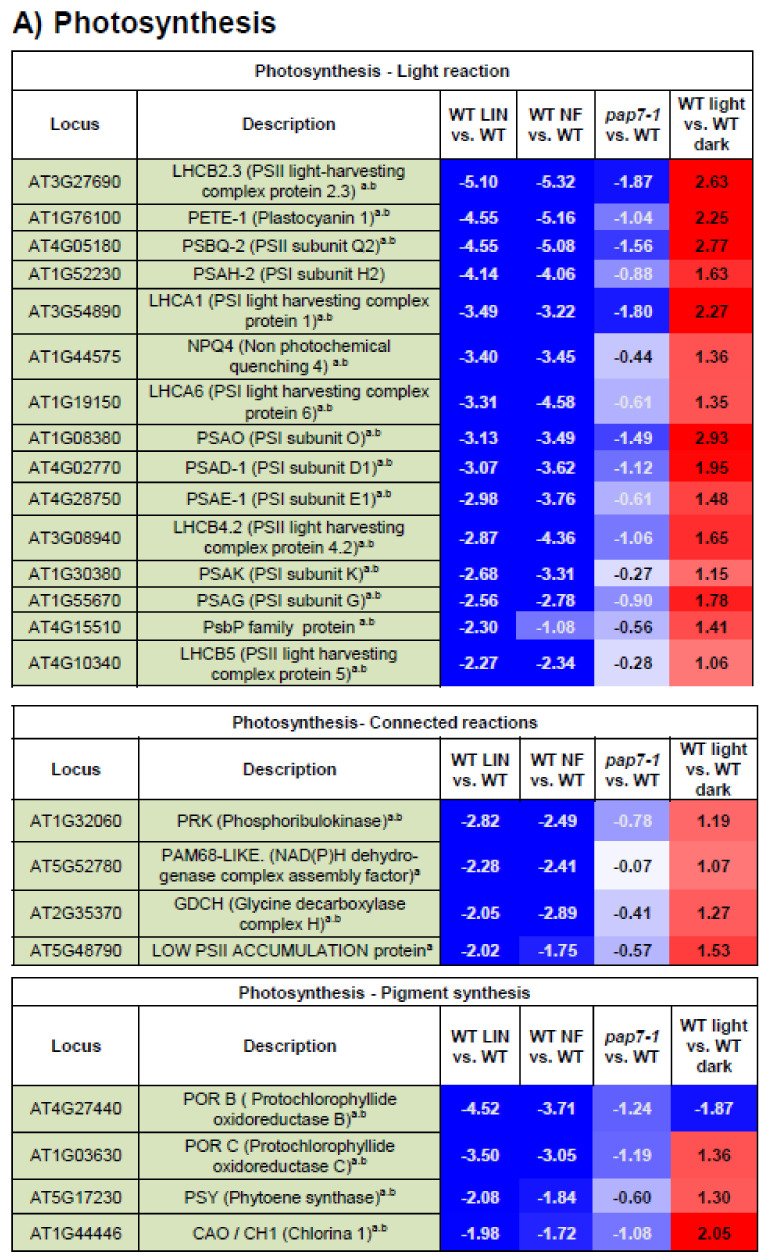
Expression patterns of selected functional groups within the identified core module. Micro-array based expression data of light-grown seedlings treated with LIN or NF are given in comparison to wild type (WT LIN vs. WT and WT NF vs. WT, respectively). Expression data from the *pap7-1* mutant are given in comparison to light-grown WT (*pap7-1* vs. WT light), and dark grown WT (*pap7-1* vs. WT dark). As general control expression data of light-grown WT in relation to dark-grown WT are given. All data represent log2-fold expression changes and are supported by colour code indicated in the bottom right corner of figure G. Gene identities (Locus) and respective encoded proteins (Description) are given in columns to the left. Functional groups are indicated on top of each table and are arranged in functionally related subsets. (**A**) Photosynthesis, (**B**) carbohydrate metabolism and transport, (**C**) redox regulation, (**D**) development, (**E**) transcription, (**F**) proteins and stress, (**G**) lipids and hormones. Genes encoding proteins with predicted or proven plastid localization are highlighted in light-green. For details see Appendix A.

**Figure 3 plants-10-00296-f003:**
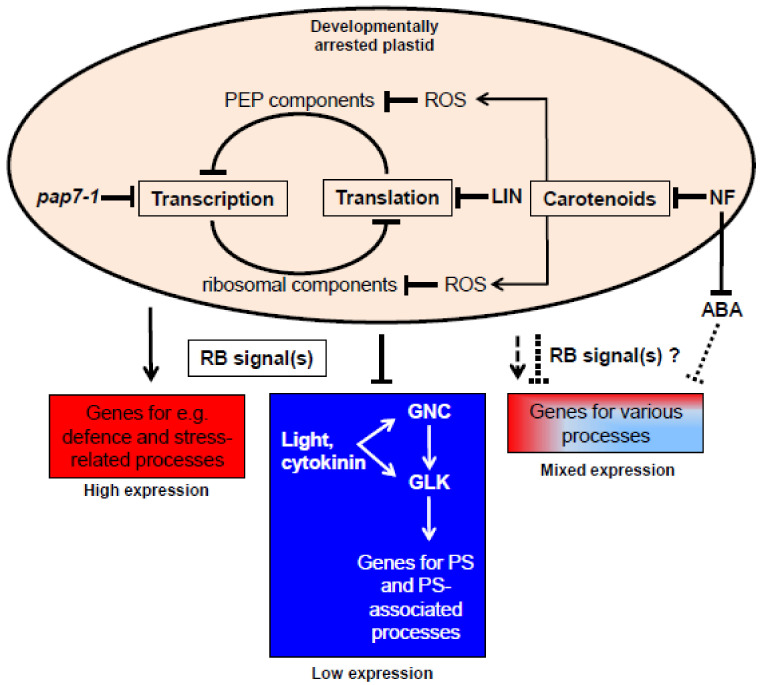
Working hypothesis for the action of retrograde biogenic signals. Oval on top represents a plastid that is arrested in its development either genetically (*pap7-1* mutant) or chemically (LIN or NF). The molecular inhibitory effects are connected by negative feedback loops resulting in a uniform retrograde biogenic (RB) signal in all three cases. Major target is the group of photosynthesis and photosynthesis-associated genes that normally are activated during chloroplast biogenesis by light and cytokinins with GNC and GLK transcription factors as major regulatory hubs. RB signals intercept into this activation resulting in low expression of these genes. RB signals may act also oppositely by promoting the expression of a small set of genes, many of them are involved in or related to defence responses to biotic and abiotic stresses. Largely not understood is the action of RB signals on genes with mixed expression. Likely additional regulatory factors may play a role such as the suppression of abscisic acid (ABA) formation upon NF treatment. Arrows represent conceptual positive influences, bars represent conceptual negative influences.

## Data Availability

Original data sources are described in the Methods section.

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
