# Peer review of "A Core Module of Nuclear Genes Regulated by Biogenic Retrograde Signals from Plastids"

_plants, 2021, doi:10.3390/plants10020296_

Round 1

Reviewer 1 Report

Manuscript ID: plants-1098018

Dear Editor, Dear Authors,

Thank you for letting me read your work.

The Manuscript “A Core Module of Nuclear Genes Targeted by Biogenic Retrograde Signals from Plastids” is interesting and present the meta-analysis of a few array data on chloroplast biogenesis. The result about a core gene-set responding specifically to chloroplast biogenesis inhibition is intriguing and can enrich the knowledge of the scientific community on the subject.

It is my opinion that sometimes Authors are too confident in their conclusions: even though a general genetic behavior is clearly recognizable from the array data, exposing models about single genes would require a confirmation by qPCR or more solid data from RNA seq analyses.

In addition, the model proposed is based on the existence of a blocked chloroplast biogenesis signal, but I did not find an adequate support for this hypothesis in the MS. Isn’t the model explained also by the lack of a normal chloroplast biogenesis signal?

Point by point suggestions:

Line 2: R*etro-grade

Line 14: why control? Pathways

Line 27: Special is not clear.

Line 28: important is too generic

Line 44: deve-lopment typo

Line 66-68: copy of genes. The number of genes is in favor of nuclear ones.

Line 127: diagram*

Legend Figure 1: Venn diagram*

Line 206: approximately*

Line 208: in the wild type (not control conditions)

Line 209: the block

Line 210: response instead of expression pattern

Line 217: remove here identified (core module is clear)

Line 220: remove expression.

Line 222: remove with respect… or make the sentence more intelligible.

Line 222: corresponding to what?

Lines 225 – 230: the comparison of genetic inhibition to pharmaceutical treatments is not correct in general. Please state clear you are speaking of the specific treatment performed.

Line 248: demonstrate is too strong. Maybe support. It is based on a prediction.

Line 256: databases*

Figure 2: Please remove the values between +/- log2=1 and not significative p=0.05 as stated in the M&M. put a 0, ND or at least a white background.

Please remove the pap7-1 vs wt dark column. It is never used, and it is not explained how the inhibition of translation in the chloroplast may induce the response in the dark.

There is a problem with the quality of the image/size in the B/C/D/E panels

It would be worth to comment that you only found GLK1 and not GLK2

Line 287: components is repeated.

Line 300: many other*

Line 307: The model here is not clear. Why a signal for the blocked development rather than the lack of a signal for the accomplished one?

Line 308: signal is repeated too many times. Please make the sentence more readable

Line 319: Many is too general. Half of the core…set. Or all the chloroplastic….

Line 323: has been detected for instead of exhibit

Line 329: genes encoding for

Line 332: component -s. etioplast*

Check gene and protein nomenclature rules. Genes should be italic and proteins start with a majuscule.

Line 335: ref

Line 339: ref

Line 342: I would remove complex. What is the meaning of complex for an expression pattern

Line 344: gene italic

Line 360: italic gene

Line 371: genes in italic

Line 378: genes in italic

Line 387: a number of studies but just one review has been reported. Please cite original articles.

Lines 389-391: Even though this is a work based on a meta-analysis, I find this sentence more adapted to a review. Please comment mainly data presented.

Line 397: functions instead of levels?

Line 399: the following article could be useful for sustaining this hypothesis: https://doi-org.insb.bib.cnrs.fr/10.1111/tpj.14700

Line 409: very complex, as before it is an expression profile. I would just leave difficult to interpret.

Line 421: reference for OPDA movements

Line 427-428

Line 441: negative feedback loops. needs ref

Line 442: initiate signal. Why not lack of signal?

Line 447: effect instead of character?

Line 454: virtually ? meaning

Line 456: target of/for

Reviewer 2 Report

I have reviewed the article entitled "A core module of nuclear genes targeted by biogenic retrograde signals from plastids" submitted to Plants. In this article, to identify the standard set of genes affected by biogenic retrograde signaling, the authors performed a meta-analysis of transcriptome data from 3 different experiments: transcription (pap7), translation (Lin), and oxidative stress (NF). Subsequently, the authors identified genes affected by all experimental conditions. The authors' idea is interesting, and the obtained gene list will provide a novel insight into this scientific field. However, I'm not convinced that the identified gene set truly represents "a core module" of genes targeted by RB signal. Although functions of some gene sets have already been established to be targetted by RB signal, it has not been demonstrated whether other novel genes have a primary target of RB signal or not. The possibility of secondary or pleiotropic effect cannot be excluded. In this sense, I would recommend toning down "a core module" in the title and throughout the text.

The followings are minor points for correction:
Title: Tetrograde -> Retrograde
Line 44: development
Line 332: Reference for SPD1
Line 332: eoplast -> etioplast
Line 335: Reference for peroxin 11
Line 338: Reference for JA production in peroxisome
Line 338: Reference for actin 8
Line 362: Reference for COL7
Supplemental Table 1: Although the reference is included, a reference list is lacking.

Reviewer 3 Report

The manuscript by Grubler et al. identified a core module of nuclear genes regulated by retrograde signals from plastids when plastid functions are defective. Authors performed meta-analysis using three expression data sets from Lincomycin treatment, norflurazon treatment, and pap7 albino mutant, which all of them causes defects of chloroplast biogenesis based on phenotype. It has been generally known that photosynthesis-associated nuclear genes (PhANGs) have been regulated by retrograde signaling. However, all those PhANGs were not repressed in all these conditions, questioning which genes are indeed targeted by retrograde signaling. This manuscript identified that there are a core group of nuclear genes regulated by all three different treatments/genetic conditions. Although chemical treatments (especially NF) should be carefully interpreted as the concentration of NF will impact the strength of the effect, a list of genes regulated by all chemical treatments and albino pap7 mutant can be recognized as a core module of genes. Authors' conclusion is well-supported by the analysis. Considering the necessity to precisely define core modules of genes affected by retrograde signal, this manuscript provides valuable information to the area of chloroplast biogenesis and retrograde signaling. I would note that those genes might be still dependent on conditions (e.g. light quality, quantity, developmental stages), which is beyond of the current scope in this manuscript. It would be interesting how those core genes are dependent or independent of retrograde signaling in different environmental conditions.